# Impact of COVID-19 Pandemic on (Health) Care Situation of People with Parkinson’s Disease in Germany (Care4PD)

**DOI:** 10.3390/brainsci12010062

**Published:** 2021-12-31

**Authors:** Odette Fründt, Anne-Marie Hanff, Tobias Mai, Christiane Kirchner, Emma Bouzanne des Mazery, Ali Amouzandeh, Carsten Buhmann, Rejko Krüger, Martin Südmeyer

**Affiliations:** 1Department of Neurology, Klinikum Ernst von Bergmann, Charlottenstraße 72, 14467 Potsdam, Germany; christiane.kirchner@klinikumevb.de (C.K.); Emma.BouzannedesMazery@klinikumevb.de (E.B.d.M.); ali.amouzandeh@klinikumevb.de (A.A.); Martin.Suedmeyer@klinikumevb.de (M.S.); 2Transversal Translational Medicine, Luxembourg Institute of Health (LIH), 1A-B, Rue Thomas Edison, L-1445 Luxembourg, Luxembourg; Anne-Marie.Hanff@lih.lu (A.-M.H.); rejko.krueger@uni.lu (R.K.); 3Department of Nursing Development/Nursing Research, University Hospital Frankfurt, Theodor-Stern-Kai 7, 60590 Frankfurt, Germany; Tobias.Mai@kgu.de; 4Department of Neurology, University Medical Center Hamburg-Eppendorf, Martinistraße 52, 20246 Hamburg, Germany; buhmann@uke.de; 5Luxembourg Centre for Systems Biomedicine (LCSB), University of Luxembourg, 6, Avenue du Swing, L-4367 Luxembourg, Luxembourg; 6Parkinson Research Clinic, Centre Hospitalier de Luxembourg (CHL), 4, Rue Nicolas Ernest Barblé, L-1210 Luxembourg, Luxembourg; 7Department of Neurology, University Medical Center Düsseldorf, Moorenstraße 5, 40225 Düsseldorf, Germany

**Keywords:** Parkinson’s disease, corona, COVID-19, care, Germany, impact, impairments, telemedicine, Care4PD, vaccination

## Abstract

The Care4PD study examined the impact of the COVID-19 pandemic on the care situation of people (PwP) with Parkinson’s disease in Germany. A comprehensive, nationwide, anonymous questionnaire for PwP was distributed by the members’ journal of the German Parkinson’s Disease Association and in several PD specialized in- and outpatient institutions. PwP subjectively evaluated their general care situation and individual impairments during the pandemic. We analyzed 1269 eligible out of 1437 returned questionnaires (88.3%) and compared PwP with (p-LTC) and without (np-LTC) professional long-term care. Both groups rated the general pandemic-related consequences as being rather mild to moderate (e.g., worsening of symptom or concerns). However, familial/social contact restrictions were indicated as most compromising, whereas access to outpatient professional health care providers was less affected. PwP with professional LTC reported more impairment than those without. COVID-19 vaccination rates and acceptance were generally high (p-LTC: 64.3%, np-LTC: 52.3%) at the time of the study, but realization of sanitary measures—especially wearing masks as a patient during care sessions—still needs to be improved. Technical options for telemedicine were principally available but only rarely used. Altogether, during the COVID-19 pandemic, PwP in Germany seemed to have a relatively stable health care access, at least in outpatient settings, while mainly social isolation compromised them. The p-LTC group was more impaired in everyday live compared with the np-LTC group.

## 1. Introduction

Data are inconclusive on whether the diagnosis of Parkinson’s Disease (PD) is a specific risk factor for a negative COVID-19 outcome [1,2,3,4,5,6,7] or whether a COVID-19 infection itself could cause a neurodegenerative disease such as PD [8,9,10,11]. The International Parkinson and Movement Disorder Society published a viewpoint with behavioral recommendations—independent of the general social restrictions to prevent the viral spread—such as strictly practicing social distancing, avoiding in-hospital stays for non-emergency reasons, postponing elective DBS surgery, and using telemedicine instead of direct outpatient visits [1]. In addition to these suggestions and the general limitations, disruptions of everyday activities and reduced access to health care and therapists might further complicate the situation of people with Parkinson’s disease (PwP) during the COVID-19 pandemic. Global and regional studies from America (US, Canada, Brazil), Asia (China, Japan, South Korea, India, Iran Israel, Turkey), North Africa (Egypt), or European countries (Luxembourg, Netherlands, UK, Italy, Spain, Slovenia) reported wide-ranging consequences of the pandemic on PwP, such as worsened motor and non-motor symptoms [12,13,14,15,16,17,18,19,20,21,22,23,24,25,26,27,28], a negative impact on mental health [29], a decline in quality of life [20,21,30], disrupted social and medical activities [12,13], impaired access to PD medication [21,31,32], or other unmet needs such as emotional distress, problems with rescheduling appointments with health care providers, and reduction in physical activity [2,14,20,22,24,33,34]. On the other side, although hours of caregiving at home and caregiver burden increased [33], patients and caregivers seem to be well informed and to have coped well in some studies [35,36,37].

To our knowledge there are currently only two studies published examining the impact of the COVID-19 pandemic on PwP in Germany, either focusing on knowledge, attitudes, and preventive behavior of PwP during the pandemic [38] or on the decline in PD multimodal complex treatment and application of pump-based therapies [39]. As part of the comprehensive, nationwide Care4PD patient survey, we here explicitly examine the impact of the COVID-19 pandemic on the real-word health care situation of PwP in general and with special focus on long-term care in Germany—a field that has only attracted little scientific attention nationally and internationally as of yet [40].

## 2. Materials and Methods

The Care4PD patient survey (available at [41]) was developed to evaluate the care situation of PwP in Germany in general (part I) and of those PwP employing professional long-term care (LTC) services (part II) such as outpatient care services, professional domestic 24-h care by external care staff or nursing homes. In addition to demographic and disease-related questions, the current care situation of PwP (e.g., use of professional care, care degree, availability of house calls, support in everyday life, etc.) and the impact of the COVID-19 pandemic on the individual care situation were investigated.

A testing phase of the initial questionnaire draft comprised interviews with ten PwP and their caregivers, with a specialist in gerontology experienced in the field of barrier-free accessibility, with several movement disorders specialists, and with a statistician. Afterwards, the questionnaire was revised, condensed, and optimized to the final version. This finally included 56 questions in total, with 13 COVID-specific questions (part I: 10, part II: 3) that are attached as Appendix A. Single or multiple-choice questions, visual analogue scales (from 0 = not applicable/not at all to 10 = very applicable/very much), and open questions were used.

Questionnaires were distributed nationwide using the members’ journal of the German Parkinson Association (Deutsche Parkinson Vereinigung e.V., dPV) and via post in several PD specialized in- and outpatient institutions, with a circulation of about 25,000 copies. Study participation was voluntary and anonymous. The study was approved by the local ethics committee (reference: S10(bB)/2021). The questionnaire was deployed and collected from March to July 2021.

Returned questionnaires (*n* = 1437, response rate about 5.7%) were scanned and automatically recorded in a database using FormPro 3.0 software by OCR Systeme GmbH [42]. The quality of this automatic database feeding was good with a rate of only 3.5 ± SD 4.8 (0–60) misreading, mainly regarding detection of handwriting in free text questions or of very small or faint ticks. Throughout, the data base was supervised, manually checked for plausibility, and later imported into IBM SPSS^®^ Statistics version 27 [43] to perform statistical analysis. Questionnaires that were sent twofold (*n* = 1), with inconsistent answers (*n* = 115) or >30% missing data (*n* = 52), were excluded from analysis.

Student’s t-test was used for group comparisons of metrical variables between PwP with (“p-LTC” group) and without professional long-term care (“np-LTC” group). Corrected *p*-values were used in cases with unequal variances. Nominal or ordinal variables were compared between groups using Pearson’s chi-squared test.

In this short communication, the main demographic and COVID-19-related results of the Care4PD survey are reported.

## 3. Results

Finally, a total of 1269 out of 1437 questionnaires (88.3%) was analyzed, including 269 PwP with (21%, p-LTC) and 1000 PwP without (79%, np-LTC) professional long-term care.

### 3.1. Demographic and Clinical Data

Demographic and clinical data are shown in Table 1 separately for both groups (p-LTC vs. np-LTC). PwP with p-LTC were mainly cared for by outpatient care services (62.1%), followed by residing in nursing homes (26.0%) and those supplied by professional 24-h care (11.9%).

Compared with np-LTC PwP, those receiving professional care were older, predominantly female, scored higher in the Hoehn and Yahr stage, indicating greater disease severity, had a higher mean care degree, and were hospitalized more often for emergency or non-emergency reasons during the pandemic. Infection rates (5.5% vs. 1.7%) but also vaccination rates against SARS-CoV-2 (at least one vaccination at the time of the study from March to July 2021: 64.3% vs. 52.3%) were higher in PwP with p-LTC compared with the np-LTC group.

Although a great number of PwP (85.5% in the p-LTC and with 92.6% even significantly more PwP in the np-LTC group) would have the technical options to perform telemedicine (internet, telephone, or both combined), utilization of telemedicine in principle was only conceivable in about 50% of PwP in both groups. During the COVID-19 pandemic, on a scale from 0 (not at all) to 10 (very frequently), the actual use of telemedicine was reported to be rather seldom (mean of 2.8 in the p-LTC and 3.1 in the np-LTC group, *p* = 0.263) in both groups. There was no significant difference in the use, willingness or technical options between PwP in less populated (<20,000 inhabitant) and densely populated areas (>20,000 inhabitants, all *p* > 0.05).

### 3.2. Comparison of the Pandemics’ General Consequences and Care-Related Impairments between p-LTC and np-LTC

General consequences of the COVID-19 pandemic on PwPs’ lives were rated on a visual scale from 0 (not applicable/not at all) to 10 (very applicable/very much; see Appendix A). Results are shown in Figure 1.

Summarizing Figure 1, the COVID-19 pandemic resulted in only mild worsening of PD symptoms and only a few phases in which PwP felt less supported during the pandemic. Concerns about the pandemic and its impact on everyday life were rated as moderate in both groups. All consequences were more pronounced in the p-LTC group, with significant group differences for symptom worsening (*p* < 0.001), decrease in support (*p* < 0.001), and overall impact on everyday life (*p* = 0.043).

Regarding specific, care-related consequences during the pandemic, a greater proportion of the np-LTC group (27.8%) reported no impairment at all compared with those with p-LTC (14.9%, *p* < 0.001, see Figure 2, left graph).

PwP of both groups who indicated negative pandemic-related impact (p-LTC: 82.5%, np-LTC: 70.2%, see right graph (b)) also reported contact limitation with family, relatives, and friends as the major impairment, which was followed by limited options to leave the house or apartment. Interestingly, reduced contacts with doctors or nursing staff were only rarely mentioned. Again, most restrictions (leaving house, less contact with families, doctors, and therapists) were significantly more pronounced in the p-LTC group. The percentage of “other” impairments was comparable in both groups, with free text answers specifying limitations such as reduced social contacts in general, decreased cultural offerings, limited access to gastronomy, less physical activity (e.g., sports or swimming groups), or restricted interchange in the PD support groups.

### 3.3. Sanitary Measures during Professional Care Sessions in the p-LTC Group

Regarding sanitary measures during professional care sessions, as can be seen in Figure 3, results show that in the p-LTC group the best implemented measure was the nursing staff wearing masks (85.1%), followed by hand hygiene (68%) and room ventilation (66.5%), whereas the PwPs’ protection measures only played a minor role, with just 23.4% of them wearing a mask.

A total of 8.6% of PwP reported to not even know about the sanitary measures applied. The term “others” was mainly specified with vaccination, measuring temperature of visitors in long-term care facilities, wearing gloves, and avoiding unnecessary contacts. Furthermore, the p-LTC group felt only moderately protected by (mean 6.9 (0–10) ± SD 2.9) and informed about (mean 5.3 (0–10) ± SD 3.5) their nursing staffs’ sanitary measures’ concepts.

## 4. Discussion

Our study analyzed the impact of the COVID-19 pandemic on the care situation of PwP with (p-LTC) and without (np-LTC) professional long-term care. Our data were derived from the comprehensive, anonymous, nationwide Care4PD patient survey (available at [41]) that—additionally—targeted examination of the general care situation of PwP in Germany, as there were only very limited data available so far [44]. In particular, people with advanced PD receiving professional LTC (e.g., outpatient care services, nursing homes) only rarely participate in clinical studies and attract less scientific attention nationally and internationally [40], which leads to only sparse information on their actual care situation.

The response rate of 5.7% of this study was within the range of previous, comparable questionnaire studies of PD clientele [45,46,47]. Interestingly, the response rate resembled that of a recent questionnaire-based study (4.7%) using the same distribution method via the members’ journal of the German Parkinson Association but addressing a completely different topic [48].

Analysis of age, residence, and Hoehn and Yahr stages revealed that a representative group of PwP was reached with a typical mean age of >65 years (comparable with [44]), in both rural and urban areas and of all disease stages, although we cannot rule out that patients with atypical parkinsonism were also included in the study [49,50,51].

According to the official care statistics of the German Federal Bureau of Statistics from 2019 [52], from a total of 4,127,605 care service recipients (“Leistungsempfänger”), about 51% manage their care themselves or with the help of relatives. In our study, the overall rate of PwP with a care degree as an indirect parameter for the number of care service recipients was of 56% (*n* = 705, both groups combined) and—consistently—about 62% (*n* = 440) of them got along without professional care.

Again, according to the care statistics [52], in the group of those recipients with an institutional care service, 41% lived in fully inpatient institutions (nursing homes) and 49% were supported by outpatient care services. In contrast, in our study, only 26.0% of PwP resided in nursing homes, whereas 62.1% received care by outpatient care services and 11.9% used outpatient 24-h care (that is not explicitly mentioned in [52]). Furthermore, 44% of the np-LTC group had a care degree and received care insurance benefits. This could rather indicate that people with p-LTC are underrepresented here, maybe due to the method of recruiting (which we do not know, as we did not specifically target PwP in LTC facilities) or their lack of autonomy in filling out the questionnaire. Another explanation could be that PD differs from other chronic diseases with the need of p-LTC in such a way that a significant number of PwP have a slow disease progression (consistent with a long mean disease duration of 10–13 years, see Table 1) and thus may not reach the highly disabling disease stages that require professional LTC, as has been speculated before [53].

However, as it is presumed that about 5–10% of PwP reside in long-term care facilities [54] with—based on our knowledge—an unknown number of those receiving outpatient care services, the percentage of 21.1% PwP receiving p-LTC in our study seems to be realistic. Nevertheless, it must be mentioned that we did not ask for the specific qualification and duties of the nursing staff and thus cannot distinguish between the different care services according to the German “Sozialgesetzbuch” V and XI, such as medication control only vs. comprehensive nursing care.

As expected, the p-LTC group was older, with higher disease severity, reported a higher hospitalization rate, and had more impairments during the pandemic.

Empirically and data-based, we hypothesized that due to the pandemic PwP might have strikingly less health care assess in the inpatient sector (e.g., decreased admissions of PwP to clinics, less access to PD multimodal complex treatment and application of pump-based therapies [39]) or DBS [55]). This hypothesis was supported by our data showing that about 80% of our PwP sample reported no hospital admissions during the last 6 months of the pandemic. It is remarkable that health care access to doctors and therapists in the outpatient sector was not relevantly affected—at least for those without professional care. This indicates that the outpatient care was maintained during the pandemic crisis as a “stable pillar”, potentially also thanks to individual but likewise professional–political efforts such as the “Central Corona Platform for neurologists and psychiatrists” [56] by the German professional associations for neurologists (“Berufsverband Deutscher Neurologen”) and psychiatrists. This online initiative aimed at ensuring the continuation of constant patient care by offering practical and educational support for resident physicians during the pandemic, such as online seminars or by providing consultation, assistance, and expert information. Additionally, the use of telemedicine was recommended by the initiative to ascertain patient care during the pandemic. Potentially, telemedicine might also be an option to better reach the p-LTC clientele [57,58,59]. However, our data indicate that although technical options were accessible in most PwP (even those with p-LTC, who additionally might not only need technical options but also care staff support to use the technology), acceptance and use of this technique was rather low in both groups. Although telemedicine approaches have been shown to be effective [60,61] and are widely recommended for PwP [16,32,62], even by the International Parkinson and Movement Disorder Society [1], our findings are congruent with previous studies documenting that even high levels of satisfaction with telemedicine did not translate into a sustained interest or use of this health care approach [2,12]. Extensive promotion of remote or virtual care modes that have been found to be as effective as in-person communication [63] or alternative offerings (e.g., conducting more house calls by medical and/or therapeutical staff, cross-sectoral approaches with PD nurse specialists as in other countries that are only rarely deployed in Germany so far [64]) should be focused on in the future.

Noteworthy, the largest impact resulted from social distancing, especially distancing from families, relatives, and friends and from feeling restricted to home/room—aspects that have been found before [14,32] and that seem to increase stress levels [24,32]. In a previous study, high perceived stress in PwP was associated with lower social support [24], and COVID-19-related stressors were associated with mental health issues especially in female, highly educated people, people with advanced PD, and those vulnerable to distancing or seeking social support [65]. Moreover, other studies documented the negative impact of the pandemic on the health and function of PwP [2,16,18,29,65].

However, others found that especially family members supported the PwP during the pandemic and took care of their unmet needs, such as shopping or picking-up medication [33]. In a Danish/Swedish study, PwP ratings and written complements even suggested that there was an improvement in health-related quality of life, with the feeling that the “pressure” was gone since the beginning of the pandemic [37]. These may be some reasons why—despite the negative impact on social contacts mentioned above—PwP in our study somehow seemed to cope relatively well with the pandemic situation, with feeling only moderate concerns, moderate impact on everyday live, and little influence on PD symptoms during the pandemic.

Furthermore, although we generally asked for the whole COVID-19 pandemic period, the study period was within the “relaxation phase” between the second and third “pandemic wave” in Germany, with the availability of vaccination and extensive sanitary measures including rapid antigen tests that suggested to slowly “regain one’s life back”. All these aspects might have spread optimism amongst PwP and might have influenced our results. In our sample, until July 2021, the overall vaccination rate of 58% (64% in those with p-LTC) was comparable with that of the general population at that time (56–62% [66]), and vaccination is still intensely recommended by movement disorders specialists [67].

Although it seems that the pandemic-related general burden has been quite compensated for in the German PD community, our survey indicates that PwP with professional LTC were significantly more affected by the pandemic compared with those without. This was true regarding not only the general pandemic consequences (especially higher COVID-19 infection rate, less support, more worsening of PD symptoms) but also with a view to specific, care-related consequences (especially less contact with beloved ones, therapists, and doctors and feeling tied to the house/apartment). Still, regarding sanitary measures, a lack of protection of self and others in this group can be recognized, and doctors and caregivers should more strictly advise their PwP to wear a mask, especially during care contacts. Nonetheless, even sanitary measures that were applied most (nursing staff wearing masks and hand hygiene) were not optimally exploited. As a limitation, we did not ask for the vaccination rate of the nursing staff (also with respect to data protection reasons), but this might also be a relevant sanitary measures factor. All in all, prospectively, the p-LTC clientele should be brought into focus more intensely.

## 5. Conclusions

The negative impact of the COVID-19 pandemic on the health care situation in PwP in Germany was not as severe as expected based on prior empirical impression and inpatient data, at least with respect to the outpatient sector and at a time slightly more than one year after pandemic onset. Main self-reported impairment was due to social or familial contact restrictions and isolation. However, our data suggest that PwP receiving professional long-term care are more impaired during the pandemic compared with those without and therefore should get more attention in the future by providing alternative strategies to better reach and care for them.

## Figures and Tables

**Figure 1 brainsci-12-00062-f001:**
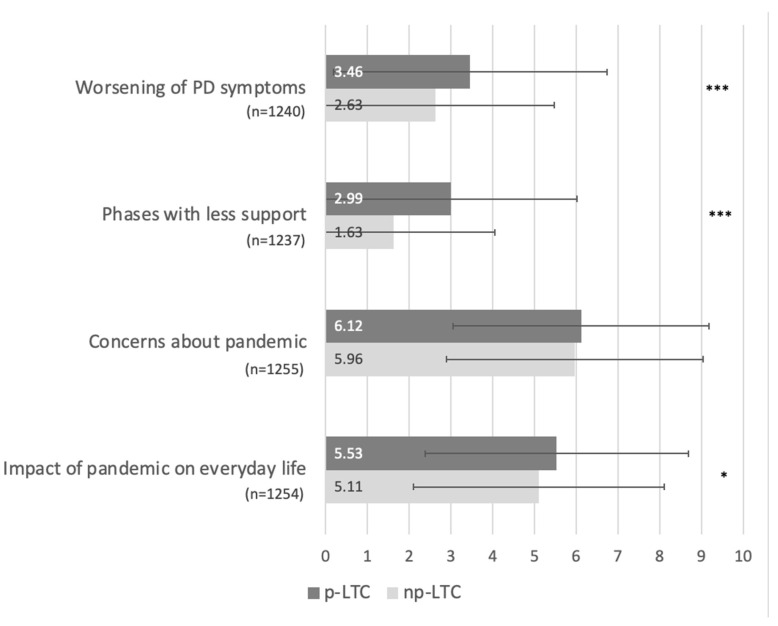
Evaluation of general consequences of COVID-19 pandemic in comparison between p-LTC and np-LTC group. People with Parkinson’s disease (PwP) evaluated the general consequences of the pandemic on everyday life on a visual scale from 0 (not applicable/not at all) to 10 (very applicable/very much). Results are depicted as mean values with standard deviation for both groups (dark grey: PwP with professional long-term care (p-LTC), light grey: PwP without long-term care (np-LTC)). Significant group differences are marked with * (* *p* < 0.05, *** *p* < 0.001).

**Figure 2 brainsci-12-00062-f002:**
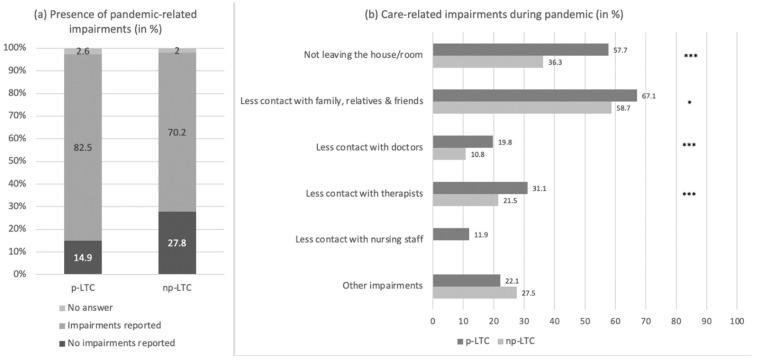
Specific care-related impairments during COVID-19 pandemic (in %). (**a**) Percentage of people with Parkinson’s disease (PwP) with or without COVID-19 pandemic-related impairments are shown for both groups with (p-LTC) and without (np-LTC) professional care. (**b**) Percentage of PwP reporting about certain care-related impairments: p-LTC = dark grey, np-LTC = light grey. Significant group differences are marked: * *p* < 0.05, *** *p* < 0.001. Multiple answers were allowed here.

**Figure 3 brainsci-12-00062-f003:**
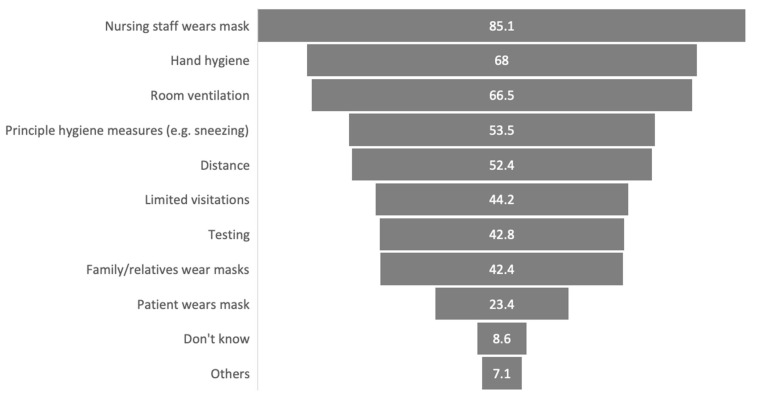
Sanitary measures during care sessions in the p-LTC group in %. Sanitary measures used during professional care sessions are listed as percentage of people with Parkinson’s disease with professional care (*n* = 269). Multiple answers were allowed here.

**Table 1 brainsci-12-00062-t001:** Demographic and clinical data of people with Parkinson’s disease with (p-LTC) and without (np-LTC) professional long-term care.

Parameter	p-LTC (*n* = 269) (Mean (Min–Max) ± SD)		np-LTC (*n* = 1000) (Mean (Min–Max) ± SD)		Statistics (*p*-Value)
Residence	<20,000 inhabitants: 39.4%	*n* = 106	<20,000 inhabitants: 39.4%	*n* = 394	*p* = 0.924
	>20,000 inhabitants: 56.1%	*n* = 151	>20,000 inhabitants: 56.9%	*n* = 569	
	(n.a.: 4.5%)	*n* = 12	(n.a.: 3.7%)	*n* = 37	
Gender	♂ 46.9%	*n* = 126	♂ 55.1%	*n* = 551	***p* = 0.013**
	♀ 52.4%	*n* = 141	♀ 43.8%	*n* = 438	
	(n.a.: 0.7%)	*n* = 2	(n.a.: 1.1%)	*n* = 11	
Age (years)	76.5 (48–95) ± SD 8.0	*n* = 265	71.5 (42–97) ± SD 8.6	*n* = 988	***p* < 0.001**
Age at diagnosis (years)	63.2 (22–87) ± SD 11.1	*n* = 255	62.0 (32–87) ± SD 10.3	*n* = 990	*p* = 0.099
Hoehn and Yahr stage	3.8 (1–5) ± SD 1.0	*n* = 238	2.8 (1–5) ± SD 1.1	*n* = 946	***p* < 0.001**
	H&Y1: 3.7%	*n* = 10	H&Y1: 18.9%	*n* = 189	
	H&Y2: 3.3%	*n* = 9	H&Y2: 9.2%	*n* = 92	
	H&Y3: 22.3%	*n* = 60	H&Y3: 43.6%	*n* = 436	
	H&Y4: 35.0%	*n* = 94	H&Y4: 19.8%	*n* = 198	
	H&Y5: 24.2%	*n* = 65	H&Y5: 3.1%	*n* = 31	
	(n.a.: 11.5%)	*n* = 31	(n.a.: 5.4%)	*n* = 54	
Care degree °	3.2 (1–5) ± SD 1.1	*n* = 269	2.5 (1–5) ± SD 0.9	*n* = 993	***p* < 0.001**
	None: 1.1%	*n* = 3	None: 54.0%	*n* = 540	
	Degree 1: 5.6%	*n* = 15	Degree 1: 6.0%	*n* = 60	
	Degree 2: 23.1%	*n* = 62	Degree 2: 16%	*n* = 160	
	Degree 3: 32.3%	*n* = 87	Degree 3: 16.1%	*n* = 161	
	Degree 4: 26.0%	*n* = 70	Degree 4: 5.0%	*n* = 50	
	Degree 5: 11.5%	*n* = 31	Degree 5: 0.9%	*n* = 9	
	Don’t know: 0.4%	*n* = 1	Don’t know: 1.3%	*n* = 13	
	(n.a.: 0%)	*n* = 0	(n.a.: 0.7%)	*n* = 7	
Hospitalization last 6 months	No admission: 75.1%	*n* = 202	No admission: 82.4%	*n* = 824	***p* = 0.023 #**
	Non-emergency: 15.6%	*n* = 42	Non-emergency: 13.5%	*n* = 135	
	Emergency: 7.1%	*n* = 19	Emergency: 3.5%	*n* = 35	
	(n.a.: 2.2%)	*n* = 6	(n.a.: 0.6%)	*n* = 6	
Telemedicine– technical options	Yes: 85.5%	*n* = 230	Yes: 92.6%	*n* = 926	***p* < 0.001 #**
	No: 11.9%	*n* = 32	No: 4.1%	*n* = 41	
	(n.a.: 2.6%)	*n* = 7	(n.a.: 3.3%)	*n* = 33	
Telemedicine– potential regular use	Yes: 48.3%	*n* = 130	Yes: 53.8%	*n* = 538	*p* = 0.094 **#**
	No: 43.1%	*n* = 116	No: 37.7%	*n* = 377	
	(n.a.: 8.6%)	*n* = 23	(n.a./multiple: 8.5%)	*n* = 85	
Proven COVID-19 Infection	Yes, with symptoms: 3.4%	*n* = 9	Yes, with symptoms: 0.8%	*n* = 8	***p* < 0.001 #**
	Yes, without symptoms: 2.2%	*n* = 6	Yes, without symptoms: 0.9%	*n* = 9	
	No: 93.7%	*n* = 252	No: 97.3%	*n* = 973	
	(n.a.: 0.7%)	*n* = 2	(n.a.: 1.0%)	*n* = 10	
Vaccination against COVID-19	Yes, already vaccinated: 64.4%	*n* = 173	Yes, already vaccinated: 52.3%	*n* = 523	***p* < 0.001** *
	Yes, I wish to: 27.1%	*n* = 73	Yes, I wish to: 40.8%	*n* = 408	
	Maybe: 4.1%	*n* = 11	Maybe: 4.0%	*n* = 40	
	No: 2.2%	*n* = 6	No: 2.3%	*n* = 23	
	(n.a.: 2.2%)	*n* = 6	(n.a.: 0.6%)	*n* = 6	

Annotations: p-LTC = patients with professional long-term care; np-LTC = patients without professional long-term care. *n* = absolute number of patients. n.a. = no or multiple answers (not included in the statistical analysis); # = Binominal comparison between those with (combined answers) and without (“no”) hospitalization or technical options, telemedicine use, or infection, respectively. * Binominal comparison between those with vaccination (combined answers “yes, I wish” or “maybe”) and those without (“no”). ° Explanation of care degrees: In Germany, the care degree is evaluated by the “Medizinischer Dienst der Krankenkassen” considering mobility, cognitive/communicative capacities, behavioral/psychiatric problems, self-sufficiency, coping with disease and therapy related issues, structuring everyday life and social contacts. Care degrees are defined as follows using a scoring system: Degree 1 = “slight impairment of independence” (score 12.5 ≤ 27), Degree 2: “substantial impairment of independence” (27 ≤ 47.5), Degree 3: “severe impairment of independence” (47.5 ≤ 70), Degree 4: “most serious impairment of independence” (70 ≤ 90), Degree 5: “most serious impairment of independence with special requirements regarding nursing care” (90–100). Patients with a care degree can apply for benefits of the nursing care insurance. Bold: statistically significant.

## Data Availability

Data are contained within this article and in Appendix A.

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
