# Peer review of "Impact of COVID-19 Pandemic on (Health) Care Situation of People with Parkinson’s Disease in Germany (Care4PD)"

_brainsci, 2021, doi:10.3390/brainsci12010062_

Round 1
Reviewer 1 Report
Authors of the study titled „Impact of COVID-19 pandemic on (health) care
situation of patients with Parkinson’s disease in Germany (Care4PD)” assess the impact of COVID-19 pandemic on the health care situation of patients with Parkinson’s disease in Germany. Certain points should be addressed:
- “Data are inconclusive whether the diagnosis of Parkinson’s Disease (PD) itself is a specific risk factor for a negative COVID-19 outcome”
Authors should add that PD, a neurodegenerative disease, possibly affected by COVID-19 may be impacted by neuroinflammation and be related to human endogenous retroviruses. ïƒ “Recent data have shown that the activation of microglia in neurodegenerative diseases and schizophrenia may be related to human endogenous retroviruses”
Ref.
- Gruchot J., Kremer D., Küry P. (2019). Neural cell responses upon exposure to human endogenous retroviruses. Front. Genet. 10:655. 10.3389/fgene.2019.0065
- One of the limitations is lack of acknowledging early onset PD patients. How in the opinion of the authors could this have impacted the results?
- As a possibly interesting parameter could be interpreted the level of education and wealth. Was is it verified?
- 4. Sanitary measures during care sessions in the p-LTC group in % -> I think that the lack of information concerning the vaccination of nursing staff is a limitation, as it may be interpreted as a possibly relevant factor
- Authors could briefly ackwoledge the issue considering the care of patients with PSP-P, which in the early stages could be misdiagnosed as PD. (-> Reviews considering PSP-P)
Reviewer 2 Report
In the manuscript entitled “Impact of COVID-19 pandemic on (health) care situation of patients with Parkinson’s disease in Germany (Care4PD)”, authors have formulated a set of questionnaires for people with Parkinson’s disease (PwP) and gathered the information from respondents across the nation. The eligible responses (1269) are divided into two groups; 1) PwP with professional long term care (pLTC) and 2) PwP with non-professional long term care (npLTC) and analyzed based on various parameters. Comparative analysis indicates that pLTC group with higher disease severity and higher hospitalization rate showed more impairments during the pandemic compared to npLTC group. The overall results of the survey showed relatively stable health care access to PwP even during the COVID-19 pandemic. This article describes the data gathered from a set of well formulated questionnaires with suitable statistical analysis, therefore, this article should be considered for publication in the journal “Brain Sciences”.
Round 2
Reviewer 1 Report
Authors have implemented most of my comments and I believe that this is a valuable work. I have one minor comment.
"in both rural and urban areas and of all disease stages although we cannot rule out that also patients with atypical parkinsonism were included in the study"
I believe adequate references highlighting overlaps between PD and atypical parkinsonisms should come after this statement. Authors can find various works on this issue.
